# Genome-Wide Analysis of the YABBY Transcription Factor Family in Rapeseed (*Brassica napus* L.)

**DOI:** 10.3390/genes12070981

**Published:** 2021-06-27

**Authors:** Jichun Xia, Dong Wang, Yuzhou Peng, Wenning Wang, Qianqian Wang, Yang Xu, Tongzhou Li, Kai Zhang, Jiana Li, Xinfu Xu

**Affiliations:** 1Chongqing Engineering Research Center for Rapeseed, College of Agronomy and Biotechnology, Southwest University, Chongqing 400715, China; x19980205@email.swu.edu.cn (J.X.); wd0823520@email.swu.edu.cn (D.W.); pyz739869777@email.swu.edu.cn (Y.P.); wwn1112@email.swu.edu.cn (W.W.); wangqianqian@email.swu.edu.cn (Q.W.); miemie2019@email.swu.edu.cn (Y.X.); ltz199896@email.swu.edu.cn (T.L.); zky12138@email.swu.edu.cn (K.Z.); ljn1950@swu.edu.cn (J.L.); 2Academy of Agricultural Sciences, Southwest University, Chongqing 400715, China

**Keywords:** *Brassica napus*, *YABBY* gene family, whole genome, phylogenetic analysis, expression pattern

## Abstract

The YABBY family of plant-specific transcription factors play important regulatory roles during the development of leaves and floral organs, but their functions in *Brassica* species are incompletely understood. Here, we identified 79 *YABBY* genes from *Arabidopsis thaliana* and five *Brassica* species (*B. rapa*, *B. nigra*, *B. oleracea*, *B. juncea*, and *B. napus*). A phylogenetic analysis of YABBY proteins separated them into five clusters (YAB1–YAB5) with representatives from all five *Brassica* species, suggesting a high degree of conservation and similar functions within each subfamily. We determined the gene structure, chromosomal location, and expression patterns of the 21 *BnaYAB* genes identified, revealing extensive duplication events and gene loss following polyploidization. Changes in exon–intron structure during evolution may have driven differentiation in expression patterns and functions, combined with purifying selection, as evidenced by *K*_a_/*K*_s_ values below 1. Based on transcriptome sequencing data, we selected nine genes with high expression at the flowering stage. qRT-PCR analysis further indicated that most *BnaYAB* family members are tissue-specific and exhibit different expression patterns in various tissues and organs of *B. napus*. This preliminary study of the characteristics of the *YABBY* gene family in the *Brassica napus* genome provides theoretical support and reference for the later functional identification of the family genes.

## 1. Introduction

The *YABBY* family is a small gene family unique to seed plants [1]. In angiosperms, *YABBY* members regulate the growth of leaves [2,3,4,5] and floral organs [6,7]. YABBY proteins belong to the zinc finger protein superfamily and are characterized by a C_2_C_2_ zinc finger domain at their N terminus and a C-terminal helix–loop–helix motif (called the YABBY domain). The C_2_C_2_ and YABBY domains are highly conserved across family members, while the rest of the YABBY proteins have low sequence similarity [5,8,9,10,11]. Phylogenetic analysis divided the angiosperm YABBY family into five subfamilies: CRABS CLAW (CRC), FILAMENTOUS FLOWER (FIL)/YABBY3 (YAB3), INNER NO OUTER (INO), YABBY2 (YAB2), and YABBY5 (YAB5) [2,12]. FIL and YAB3 are highly similar and are expressed in the abaxial side of the initial primordium, which determines the fate of abaxial cells. FIL and YAB3 may be derived from a gene duplication event and are therefore included in the same subfamily [6,13,14]. All five subfamilies are represented in early-diverging Amborellales, Nymphaeales, and Austrobaileyales (ANA) angiosperms, indicating that the last common ancestor of current flowering plants harbored at least five *YABBY* genes [7,15,16]. 

YABBY members exist in many species, but the family is small. The Arabidopsis (*Arabidopsis thaliana*) *YABBY* gene family consists of six members [3,5,17,18]. FIL, YAB2, YAB3, and YAB5 regulate gene expression in vegetative tissues [19,20], including the polar development of lateral organs, the formation of edges, the maturity of leaves, and the development of shoot apical meristems and phyllodes. FIL is expressed in both leaf primordia and floral organs [3,21,22]. As a transcriptional regulator, FIL not only is involved in the formation of floral organs and leaf development [17,23], but also affects anthocyanin accumulation [24]. INO and CRC genes are more tissue-specific: INO plays an important part in the development of the outer integument [18], while CRC was the first gene identified as having a role in the formation of nectaries in flowers at the base of stamen filaments in Arabidopsis [5,13]. The number of *YABBY* genes has been determined in multiple plant species: 9 *YABBY* genes in tomato (*Solanum lycopersicum*) that map to 7 of the 12 tomato chromosomes [25], 8 in rice (*Oryza sativa*) that form four subgroups [4], 12 in Chinese cabbage (*Brassica rapa*) mapping to 6 of its 20 chromosomes [26], and 13 in maize (*Zea mays*) [27]. The genomes of tree cotton (*Gossypium arboreum*) and *Gossypium raymondii* each encode 12 *YABBY* genes, while that of upland cotton (*Gossypium hirsutum*) harbors 23 *YABBY* genes [28]. Fifty-five *YABBY* genes were identified in seven Magnolia species, comprising 5 INO, 6 CRC, 8 YAB2, 22 YAB5, and 14 FIL members [29]. Eight orchid species have 54 *YABBY* genes separated into 15 CRC/DL, 8 INO, 17 YAB2, and 14 FIL members [30]. As the terrestrial plants bryophytes and stone pine (*Pinus pinea*) have no YABBY members, it is thought that this gene family is specific to seed plants [16].

The *Brassica* genus includes the diploid species *B. rapa*, *Brassica oleracea* (CC, 2*n* = 18), and black mustard (*Brassica nigra*) and the amphidiploid species rapeseed (*Brassica napus*), brown mustard (*Brassica juncea*), and Ethiopian mustard (*Brassica carinata*). *B. rapa*, *B. oleracea*, and *B. nigra* are diploid species that can generate tetraploid species through mutual hybridization and natural chromosomal doubling [31]. These species provide an excellent evolutionary model for the expansion of the *Brassica* genus [32]. Rapeseed (*B. napus*) is a very important oil crop worldwide and has growth habits similar to those of the model plant Arabidopsis. Rapeseed oil has high nutritional value, and its stems and leaves can be used as animal feed and for some industrial uses [33]. Although the *YABBY* gene family has been described in Arabidopsis and other plants [3,5,17,18], it remains to be characterized in *B. napus*.

In this study, using whole-genome analysis, we explored the evolutionary relationship of *YABBY* genes by systematically identifying *YABBY* genes in five *Brassica* species. We then conducted a phylogenetic and gene structure analysis of the resulting 79 genes from these five species and Arabidopsis. Focusing on *YABBY* genes in *B. napus*, we listed all *cis*-acting elements in their promoters and identified differentially expressed genes in floral organs, seeds, and siliques via transcriptome deep sequencing (RNA-seq) data from the rapeseed cultivar ZS11, which we validated by RT-qPCR. Our results provide theoretical support and reference basis for the functional identification of the *YABBY* genes in *B. napus*.

## 2. Materials and Methods

### 2.1. Identification of YABBY Family Genes in Brassica

Arabidopsis YABBY protein sequences were downloaded from the TAIR10 database (ftp://ftp.Arabidopsis.org (accessed on 25 January 2021)) and used as queries for the Basic Local Alignment Sequence Tool for Protein (BLASTP) [34] to search the predicted proteomes from five *Brassica* species on the *Brassica* Database (BRAD; http://brassicadb.cn/ (accessed on 25 January 2021)) website with *E*-value < 1 × 10^−20^. The physicochemical properties of predicted YABBY proteins, such as isoelectric points (pI) and predicted molecular weights (MW), were determined with the ProtParam tool on the ExPASy server [35] (https://web.expasy.org/protparam/ (accessed on 3 March 2021)).

### 2.2. Phylogenetic Analysis

A multiple sequence alignment and phylogenetic analysis were used to explore the evolutionary relationship among the YABBY proteins from the six species under study. The phylogenetic tree was generated in MEGA 7.0 (Tokyo Metropolitan University, Tokyo, Japan) [36] using the neighbor-joining (NJ) method with default parameters for multiple sequence alignment, with a bootstrap number set to 1000 to estimate branch lengths. The tree was visualized with Evolview (https://evolgenius.info//evolview-v2/ (accessed on 2 February 2021)) software.

### 2.3. Gene Structure, Prediction of Conserved Sequences, and Protein Domain Analysis

For each *YABBY* gene obtained from the five *Brassica* species, the corresponding annotation was extracted from the GFF file of the appropriate genome and provided as input to the gene structure display server (GSDS v2.0; http://gsds.gao-lab.org/ (accessed on 29 January 2021)) to analyze gene structure. The online tool Multiple Expectation Maximization for Motif Elucidation suite (MEME v4.12.0, http://meme-suite.org/tools/meme (accessed on 28 January 2021)) [37] was used to predict conservative structural elements; the number of motifs was set to 10, and the width was set to 6–300 each motif with an *E*-value < 1 × 10^−10^ was retained. Conserved protein domains were identified through the National Center for Biotechnological Information Conserved Domain Database (NCBI-CDD; https://www.ncbi.nlm.nih.gov/Structure/bwrpsb/bwrpsb.cgi (accessed on 29 January 2021)) [38]. The results of gene structure and protein domain were displayed using TBtools [39].

### 2.4. Chromosomal Location and Collinearity Analysis

Using the information contained in the GFF annotation file of the *B**. napus* genome, all *YABBY* genes were mapped to their respective chromosomes, and their locations were visualized with TBtools. The MCScanX algorithm [40] was used to predict collinearity between genomes, and Circos was used for visualization [41].

### 2.5. Analysis of Selection Pressure and Cis-Regulatory Elements 

TBtools was used to calculate the nonsynonymous substitution rate (*K*_a_), synonymous substitution rate (*K*_s_), and *K*_a_/*K*_s_ ratio for each pair of duplicated genes in *B**. napus* with default parameter settings. Promoter sequences (2000 bp of sequence upstream of the transcription start site) for *BnaYABBY* genes were downloaded from the NCBI database (https://blast.ncbi.nlm.nih.gov/ (accessed on 30 January 2021)) and scanned for *cis*-regulatory elements on the New PLACE database (https://www.dna.affrc.go.jp/PLACE/?action=newpLACE (accessed on 30 January 2021)) [42]. TBtools was used to draw a heatmap to visualize the number of *cis*-elements.

### 2.6. Tissue Expression Analysis

The transcriptome data (PRJNA358784) of 49 tissues and organs of ‘ZS11’ in different developmental stages were selected to analyze the expression of *YABBY*. The log_2_ FPKM was calculated for each gene and sample and visualized with TBtools and HeatMap.

### 2.7. Plant Materials

Seeds for the *B. napus* cultivar ZS11 were obtained from the Rapeseed Engineering Research Center of Southwest University in Chongqing, China (CERCR). The plants were grown in the field in Chongqing. Samples were collected at different growth stages (initial flowering stage, full flowering stage, and green pod stage) and from various tissues (leaves, petals, stamens, nectaries, seeds, siliques) and immediately frozen in liquid nitrogen and stored at −80 °C until use.

### 2.8. Total RNA Extraction and RT-qPCR Analysis

Total RNA was extracted from collected samples with the DNAaway RNA Mini-prep Kit (Sangon Biotech, Shanghai, China) and used as a template for first-strand cDNA synthesis with TransScript One-Step gDNA Removal and cDNA Synthesis SuperMix (TransGen Biotech, Beijing, China). qPCR was performed using the ChamQ Universal SYBR qPCR Master Mix (Vazyme Biotech, Nanjing, China) on a Bio-Rad CFX96 Real-Time System (Bio-Rad Laboratories, Hercules, CA, USA) as previously described [43]. *BnACTIN7* (EV116054) was employed as a reference gene and determined by the 2^−∆∆Ct^ method [44,45]. The experiment was repeated three times, and the values represent the mean ± standard error (SE). The qRT-PCR primers were obtained from the qPCR Primer Database (Appendix A) [46]. GraphPad Prism 5.0 software was used to visualize the results [47,48].

## 3. Results

### 3.1. Identification and Evolutionary Relationships of YABBY Genes

Using the six Arabidopsis YABBY proteins [2,5,49] as query, we performed a BLASTP search for related proteins across the *Brassica* genus, leading to the identification of 10 putative YABBY members from *B. rapa*, 11 from *B. oleracea*, 12 from *B. nigra*, 19 from *B. juncea*, and 21 from *B. napus* (Table 1 and Appendix A). Following the classification of the *YABBY* family in Arabidopsis, phylogenetic analysis clustered the 79 YABBY members into five subfamilies, YAB1, YAB2, YAB3, YAB4, and YAB5 (Figure 1 and Appendix A). Arabidopsis *YAB1* (*FIL*) and *YAB3* are closely related and defined one subgroup comprising the largest proportion of family members, with 29 (or 36.7%) YABBY proteins. The YAB2 subgroup contained 19 members, the YAB3 subgroup contained 15 members, and the YAB3 (which includes Arabidopsis CRC) and YAB5 subgroups were the smallest, with 8 members apiece.

The encoded proteins were 158 to 247 amino acids in length, with a predicted molecular weight (MW) ranging from 17.9 and 27.8 kDa and a pI between 5.5 and 9.95. The coding sequences of members of the *YAB1* and *YAB2* subgroups were on average longer than for the other subgroups: the coding sequences of *BnaYAB1* were between 1806 and 5159 bp, while those of *Bna**YAB2* ranged from 1375 to 5795 bp. By contrast, the coding sequence of *BnaYAB3* was 1594 bp, and those of the four *BnaYAB4* members were between 1663 and 1978 bp; the coding sequences of the two *BnaYAB5* members were 2933 and 3086 bp (Appendix A).

### 3.2. Chromosomal Locations of BnaYAB Genes and Duplication Analysis

We next extracted the chromosomal coordinates of each *BnaYAB* gene from the GFF files downloaded from the BRAD. Accordingly, we mapped 18 *BnaYAB* genes to the 12 current *B. napus* chromosomes, with the remaining 3 *BnaYAB* genes mapping to the pseudochromosome Ann. Chromosomes A03, A07, A09, C03, C05, and C08 each harbored two *YABBY* genes, while chromosomes A06, A07-random, A08, C04, C06, and C07 each carried a single *YABBY* gene (Figure 2A).

Partial and tandem gene duplications are important in the generation of new gene functions and the expansion of gene families [50]. Therefore, we performed a collinearity analysis on the *YABBY* family between Arabidopsis and three of the five *Brassica* species. The syntenic relationships between Arabidopsis, *B.*
*oleracea*, and *B. rapa*; between *B. napus*, *B. oleracea*, and *B. rapa*; and between the 21 *BnaYAB* genes are shown in Figure 2B–D, respectively. We identified pairs of homologs between the *Brassica* species: 9 between Arabidopsis and *B.*
*oleracea*, 8 between Arabidopsis and *B. rapa*, 26 between *B. napus* and *B. oleracea*, and 28 between *B. napus* and *B. rapa*. We also grouped *YABBY* genes in *B. napus* into 18 gene pairs. The copy numbers of *YABBY* genes varied from one to three in Arabidopsis and *B.*
*rapa* (Table 1, Figure 2B) and from one to eight in *B**. oleracea* and *B. napus* (Table 1, Figure 2C), indicating that several gene copies may have been lost or duplicated during evolution in various cases. For example, putative homologs to Arabidopsis *YAB3* were present only in *B**. oleracea* and appeared to have been lost in *B.*
*rapa*, whereas Arabidopsis *FIL* had two putative homologs in *B**. oleracea* and one in *B.*
*rapa* (Figure 2B).

We estimated the *K*_a_, *K*_s_, and *K*_a_/*K*_s_ ratio of homologous genes in *B. napus* (Appendix A): their *K*_a_/*K*_s_ ratios were all less than 1. These results suggested that the *BnaYAB* gene family underwent purifying selection after duplication.

### 3.3. Structures of BnaYAB Genes, Conserved Motifs, and BnaYAB Protein Domain Analyses

Gene structure reflects the evolution of a gene family. We therefore compared the coding sequences (CDS) and genomic sequences of all *BnaYAB* loci to analyze their exon/intron structures and displayed the results alongside the phylogenetic tree (Figure 3). The number of exons varied from four to eight. For example, members of the *YAB1* subfamily harbored four to eight exons, as compared to five or six for *YAB2*, seven for *YAB3* and *YAB4*, and six for *YAB5* subfamily members (Figure 3 and Appendix A). *BnaYAB* genes belonging to the same subgroup thus shared highly similar exon/intron characteristics, while those from different subgroups had more varied structures.

Reflecting the phylogenetic analysis, prediction of functional motifs showed that members of each subgroup share the same conserved motifs, implying that they may have similar functions. YAB1 subfamily members were characterized by motifs 2 and 7, which were the only motifs identified in BnaAnng18520D. YAB2 subfamily members typically harbored motifs 1, 2, 4, 5, and 9, with motif 2 at the N terminus and motif 5 at the C terminus, although BnaA08g26920D appeared to lack motif 2. The one YAB3 subfamily member contained only motifs 1, 2, and 9, while YAB4 subfamily members all had the same set of motifs (motifs 1, 2, 4, 6, 8, 9, and 10). YAB5 subfamily members had motifs 1, 2, and 4 (Figure 4A). We also scanned YABBY proteins in the NCBI conserved domain database [38]. In addition to the YABBY superfamily domain (which was a prerequisite during our identification of YABBY members), most proteins contained the HMG-box superfamily domain, with the exception of BnaAnng18520D and BnaAnng40070D. The members of the YAB4 and YAB5 subfamilies also had a coiled-coil domain-containing protein 124 (Ccdc124) superfamily domain (Figure 4B).

### 3.4. Cis-Regulatory Elements in BnaYAB Promoters

To explore the possible regulatory mechanisms of *BnaYAB* gene expression in abiotic or biotic stress responses, we identified 15 different cis-regulatory elements which can be classified into four types. First, we identified phytohormone response elements, such as those for abscisic acid (ABRE), gibberellin (P-box), and methyl jasmonate (CGTCA-motif and TGACG-motif). We also detected stress-responsive elements, including those related to drought (MBS), defense and adversity (TC-rich repeats), and low temperature (LTR). A third type of *cis*-elements consisted of several light-responsive elements: AE-box, Box 4, GATA-motif, G-box, GT1-motif, MRE, and TCT-motif. Finally, we noted the presence of anaerobic induction necessary (ARE) elements (Figure 5 and Appendix A). *BnaYAB* genes are therefore likely to participate in a number of physiological and biochemical functions, such as adversity response, phytohormone pathways, and light signaling. The complement of *cis*-regulatory elements was specific to each *BnaYAB* gene, indicating that different members of the *YABBY* gene family may help regulate different aspects of plant development.

### 3.5. Transcriptional Patterns of BnaYAB Genes

We determined the expression pattern of YABBY genes by analyzing a public transcriptome dataset of *B. napus* cultivar ZS11, including roots, stems, young leaves, old leaves, buds, flower stalks, sepals, petals, unpollinated pistils, pistils, stamens, anthers, filaments, seeds, and siliques (Appendix A). The expression patterns of most *BnaYAB* genes are tissue-specific (Figure 6 and Appendix A). For example, *BnaYAB* genes are seldom expressed in rhizomes, while most *BnaYAB1* and *BnaYAB4* subfamily members showed little or no expression in each tissue over the course of their development. *BnaYAB2*, *BnaYAB3*, and *BnaYAB5* were expressed in various tissues at the flowering stage, indicating that they may regulate the development of flowers. With the exception of BnaAnng18530D, which is highly expressed in various tissues during flowering, the remaining *BnaYAB1* members were barely expressed in most tissues (Figure 6 and Appendix A). Most members of the *BnaYAB2* subfamily shared the same expression pattern, with high expression in flower tissues and siliques. The two *BnaYAB5* members, BnaA03g22670D and BnaC03g26690D, also exhibited the same expression pattern. The expression patterns of *BnaYAB* genes were thus largely consistent with their locations along the phylogenetic tree (Figure 1). These results indicate that most *BnaYAB* genes are tissue-specific and that related genes exhibit similar expression patterns (Figure 1 and Figure 6).

### 3.6. Gene Expression Analysis in Brassica napus L.

To clarify the tissue expression characteristics of *YABBY* genes, we selected nine genes with significant expression differences for RT-qPCR analysis (Appendix A), and the results were consistent with RNA-seq (Figure 7).

During the early flowering stage (Figure 7A) BnaAnng18530D, BnaA06g04870D, BnaA08g26920D, BnaA09g48870D, and BnaC08g13560D were highly expressed in petals, with little or no expression in other tissues. In addition, BnaC08g43150D was expressed only in petals, BnaA03C0367026690 was expressed in petals and nectaries, and BnaA07g27740D was expressed at relatively high levels only in nectaries. Most genes displayed the same expression pattern during the full blooming stage (Figure 7B) as during the initial flowering stage, albeit at slightly lower levels. *BnaYAB1* and *BnaYAB2* subfamily members also showed expression in sepals, with BnaC08g13560D having the highest expression in this tissue, exceeding that seen in petals. During the days after flowering (DAF) stage (Figure 7C), all *BnaYAB* genes were expressed at low levels, if at all, in seed coats, seeds, and siliques 1 day after flowering. *BnaYAB1* and *BnaYAB2* members were expressed in leaves during the green pod stage, but at low levels. These results demonstrated that most *BnaYAB* genes are tissue-specific and that *BnaYAB* transcription is differentially regulated in different organs. In addition, we observed that *BnaYAB* transcription decreases as *B. napus* flowers develop.

## 4. Discussion

*B**rassica napus*, one of the most important oil crops in the world, resulted from the hybridization of *B**. rapa* and *B**. oleracea* followed by natural chromosome doubling [51,52,53]. The availability of the genome sequences for *Brassica* crops such as *B**. rapa*, *B**. oleracea*, and *B**. napus* [54,55,56] has paved the way for the systematic analysis of several gene families encoding transcription factors such as APETALA2 (AP2)/ERF, basic helix–loop–helix (bHLH), GRAS, CONSTANS-like (COL), and WRKY [57,58,59,60,61,62]. Here, we embarked on a comparative genomics study of the *YABBY* gene family in *B**. napus*, other *Brassica* species, and the related model plant Arabidopsis. YABBY proteins are a family of transcription factors unique to seed plants that regulate the development of lateral organs and play an important role in the differentiation of the plant dorsal axis, floral organ development, and phytohormone responses [15,25,63]. Their characterization would therefore contribute to our understanding of organ formation, development, and differentiation, in particular that of floral organs, which could offer a means of increasing yield in *B. napus*.

Each Arabidopsis *YABBY* gene should have three homologs in the genomes of *B**. rapa* and *B**. oleracea* and six copies in the *B**. napus* genome, based on their evolutionary histories [55,56,64]. Starting with 6 *YABBY* genes in Arabidopsis, we identified 21 genes in *B. napus*, 10 in *B. rapa*, 11 in *B.*
*oleracea*, 12 in *B. nigra*, and 19 in *B. juncea*, numbers that do not agree with expectations. However, each Arabidopsis *YABBY* gene had one to six putative homologs in *B**. napus* and one to three putative homologs in *B**. rapa* and *B**. oleracea* (Table 1, Figure 2B–D), possibly reflecting gene loss or duplications that might be expected following hybridization and whole-genome duplication [64,65,66]. It is speculated that the *YABBY* genes have undergone strong selection during evolution, and the retained genes should have important functions in *B. napus*. Most *YABBY* genes have more than one orthologous gene in *B. napus*, which indicates that the *YABBY* gene family has expanded. However, only a few *AtYABBY* genes have fewer than six orthologous genes in *B. napus*, which is less than expected and suggests that the *YABBY* gene family has shrunk during the diversification of *B. napus*. Notably, the number of *YABBY* genes in *B**. napus* is exactly the sum of the numbers of *YABBY* genes in *B. rapa* and *B. oleracea*, indicating that there was no gene loss during this evolutionary process and pointing to a loss of *YABBY* genes in the Arabidopsis lineage. The evolution of gene families, such as the *YABBY* family here, is thus complex and varies with the family under consideration [67,68], warranting separate study in each case. A phylogenetic analysis of YABBY proteins from Arabidopsis and five *Brassica* species separated all members into five subfamilies (Figure 1). Notably, the gene redundancy patterns of their constituent members are not the same.

We analyzed the results of gene structure and conserved protein motifs and found that genes clustered in the same subfamily have similar characteristics. For example, members of the *BnaYAB4* subfamily all contained seven exons (Figure 3). Most BnaYAB members also displayed motifs 1 and 2, whereas motif 5 was specific to the BnaYAB5 subfamily and motifs 6, 8, and 10 were specific to the BnaYAB4 subfamily; most BnaYAB1 subfamily members carried motifs 3 and 7 (Figure 4A). Of the 21 *B**. napus* YABBY proteins, 19 harbored an HMG-box, a domain that is also present in HIGH MOBILITY GROUP A (HMGA, At1g14900), which modulates flowering in Arabidopsis [69]. In addition, BnaYAB4 and BnaYAB5 members also had a Ccdc124 domain (Figure 4B). Ccdc124 is conserved across eukaryotes, contains a coiled-coil domain (CCD) found in most centrosome proteins, and participates in cell division [70], hinting that an analysis of the transcriptional regulation of these two subfamilies might illuminate their role in plant cell division. The expression levels of members of the *BnaYAB4* and *BnaYAB5* subfamilies were much lower than those of the members of the other three subfamilies (Figure 6). The above conclusions indicate that gene redundancy and gene elimination occurred during the evolution of the *B**. napus YABBY* gene family, contributing to its diversification [71].

*Cis*-regulatory elements are the targets of transcription factors and thus partially determine gene expression patterns [72]. The promoters of this gene family mainly contain phytohormone response elements, such as those for abscisic acid (ABRE); stress response elements, such as drought (MBS); and light response elements, such as AE-box and anaerobic inducible elements (ARE) (Figure 5). The numerous light response elements suggest that the expression of *YABBY* genes in *B**. napus* might be tightly controlled by photosynthesis. All promoters of genes in the *BnaYAB4* subfamily contained ABREs, which mediate transcriptional regulation in response to abscisic acid [73]. This plant hormone is typically involved in abiotic stress responses and participates in the metabolic regulation of drought stress [74], stomatal closure, and the regulation of gene expression [75]. The *B**. napus* genome thus harbors more *YABBY* genes than that of Arabidopsis, with highly complex gene structures, which may explain the high variability and tolerance of these plants under different conditions. In addition, the *cis*-regulatory elements identified in their promoters shape their participation in plant growth and development, offering new directions for further research on *BnaYAB* genes.

Arabidopsis *FIL*, *YAB2*, and *YAB5* are expressed in leaves, cotyledons, and floral organs and can control the growth of side branches [3,76,77]. We were interested in *BnaYAB* genes that regulate the development of floral organs and cotyledons; therefore, we analyzed public transcriptome data for *B. napus* (Appendix A, Figure 6) and discovered that many family members are highly expressed during the flowering stage and in floral tissues, resulting in a list of nine differentially expressed genes from subfamilies *BnaYAB1-BnaYAB3* and *BnaYAB5*. Arabidopsis YABBY4 (also named INNER NO OUTER (INO)) and SUPERMAN (SUP) regulate the asymmetric growth of the outer skin of bitegmic ovules [78,79], with *INO* expressed only in the outermost cell layer of the outer envelope and promoting the growth of the outer envelope. *INO* expression is tissue-specific [18,80], a feature that is supported by its low expression levels in tissues and cotyledons at the flowering stage. We validated the differential expression of the selected genes by RT-qPCR. RNA-seq analysis showed that *BnaYAB* genes have similar expression patterns in tissues of different periods. For example, BnaAnng18530D, BnaA06g04870D, BnaA08g26920D, BnaA09g48870D, BnaC08g13560D, and BnaC08g43150D were highly expressed in petals at the initial flowering stage, while BnaA07g27740D from the CRC subfamily was expressed only in nectaries. BnaA03g22670D and BnaC03g26690D were also highly expressed in nectaries, suggesting that they participate in nectary formation [15,55]. The expression levels of the above-mentioned genes decreased at the full blooming stage and were low in all tissues during the green pod stage (Figure 7), indicating that they play different roles at different stages, thus validating the goals of this study, although the specific underlying regulatory mechanisms should be studied in more detail.

## 5. Conclusions

In this study, we conducted a systematic exploration of the *YABBY* gene family and identified 79 *YABBY* genes from Arabidopsis and five *Brassica* species (*B. rapa*, *B. nigra*, *B. oleracea*, *B. napus*, and *B. juncea*) that belonged to the previously described subgroups *YAB1*, *YAB2*, *YAB3*, *YAB4*, and *YAB5*. We analyzed the chromosomal location, gene structure, and expression patterns of the *B. napus YABBY* genes and the conserved domains and evolutionary relationship of the YABBY proteins. During polyploidization, the *YABBY* gene family underwent tandem duplications and gene loss, with evidence of purifying selection. Exon–intron structural changes may have led to changes in coding regions and affected gene expression patterns and protein functions. Finally, we combined RNA-seq and RT-qPCR analysis to explore the specific expression of *BnaYAB* genes. The results of this study offer guiding principles for the evolution of the *YABBY* gene family, providing a basis for polyploid analysis and laying theoretical support for further research on the function of *BnaYAB* genes.

## Figures and Tables

**Figure 1 genes-12-00981-f001:**
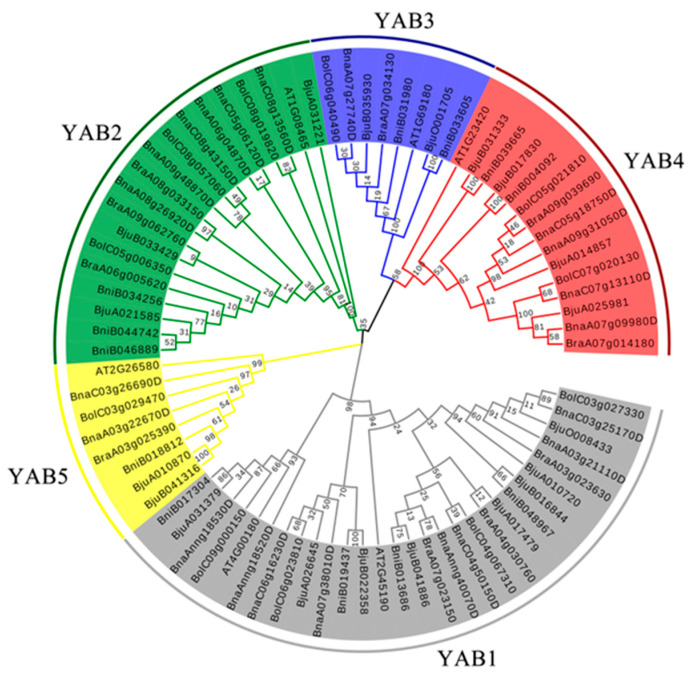
Phylogenetic tree of YABBY proteins from Arabidopsis and five Brassica species. YABBYs cluster into five subfamilies (YAB1–YAB5), indicated by different colors. Bju, Brassica juncea; Bna, Brassica napus; Bni, Brassica niger; Bol, Brassica oleracea; Bra, Brassica rapa.

**Figure 2 genes-12-00981-f002:**
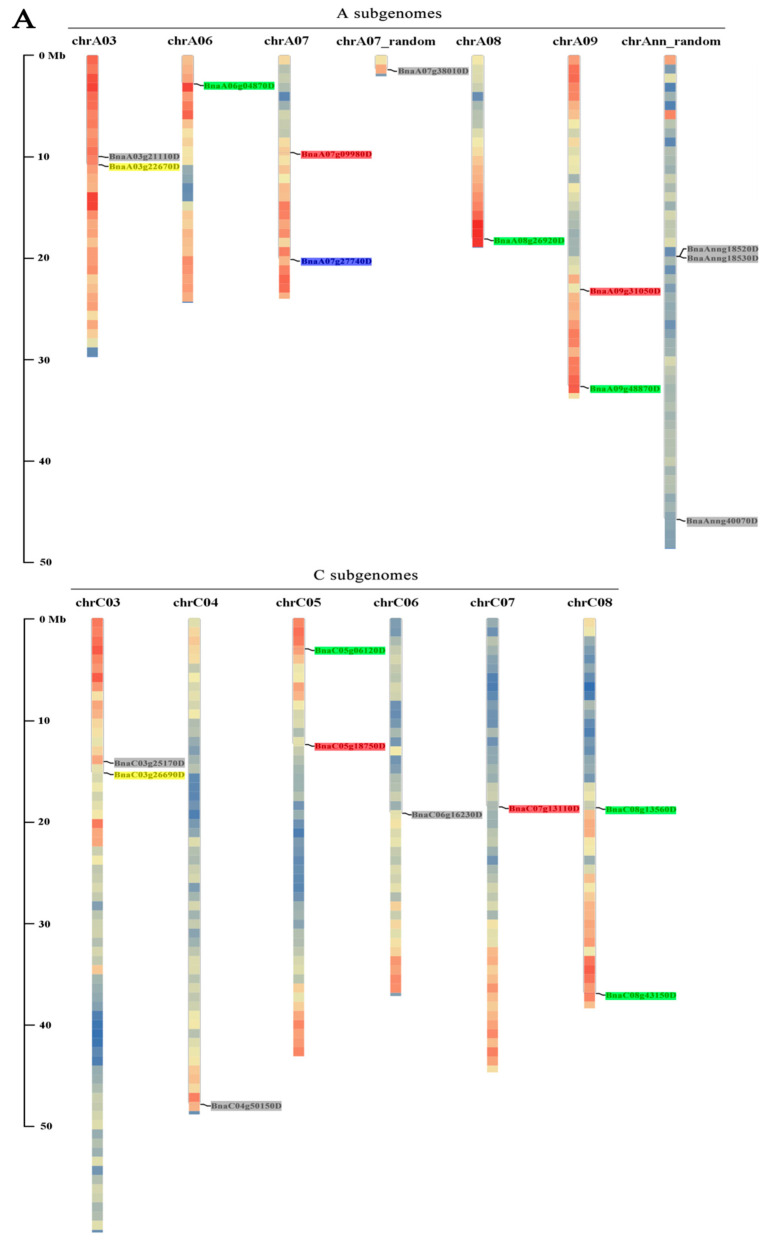
Chromosomal locations of BnaYAB genes in the *B. napus* genome and syntenic analysis of YABBY genes in Arabidopsis, *B. napus*, *B. rapa*, and *B. oleracea*. (**A**) Chromosomal location of the 21 BnaYAB genes in the *B. napus* genome. Genes from the same subgroup are highlighted in the same color, as defined in the phylogenetic tree (Figure 1). Each chromosome is drawn as a heatmap of gene density. (**B**) Collinearity analysis of YABBY family genes between Arabidopsis, *B. rapa*, and *B. oleracea*, indicated as connecting lines. (**C**) Collinearity analysis of YABBY family genes between *B. napus*, *B. rapa*, and *B. oleracea*, indicated as connecting lines. (**D**) Chromosomal locations of the 21 BnaYAB genes in the *B. napus* genome, indicated as connecting lines. A and C denote the two main subgenomes in *B. napus*. Mb, megabase; Ann, pseudomolecule chromosomes. The 5 Arabidopsis chromosomes (1–5), 19 *B. napus* chromosomes (A01–A10 and C01–C09), 10 *B. rapa* chromosomes (A01–A10), and 9 *B. oleracea* chromosomes (C01–C09) are shown. Gene pairs are represented by red lines.

**Figure 3 genes-12-00981-f003:**
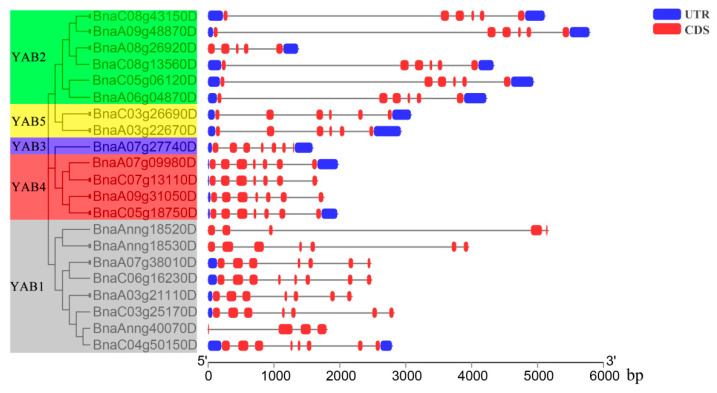
Exon–intron structures of the 21 BnaYAB genes, ordered based on their phylogenetic positions.

**Figure 4 genes-12-00981-f004:**
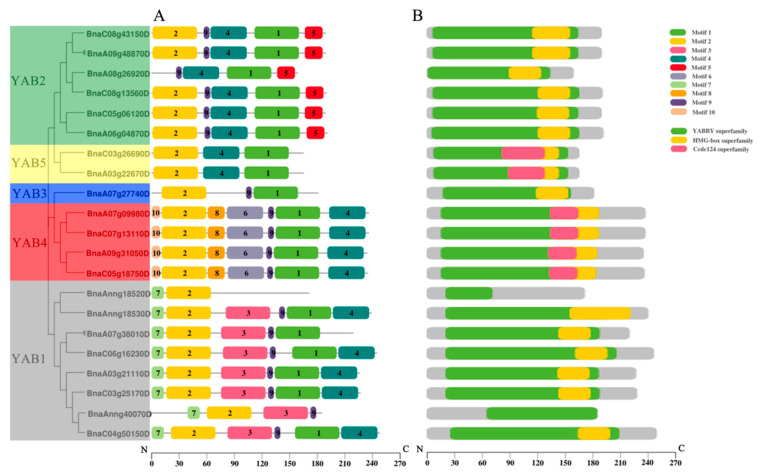
Analysis of protein motifs and conserved domains in *B. napus* YABBY proteins. (**A**) Protein motifs. Conserved motifs in the 21 BnaYAB proteins, arranged based on their phylogenetic positions. Weblogo plots of the ten conserved motifs are shown in Appendix A. (**B**) Conserved domains. Conserved domains are represented by different colored boxes.

**Figure 5 genes-12-00981-f005:**
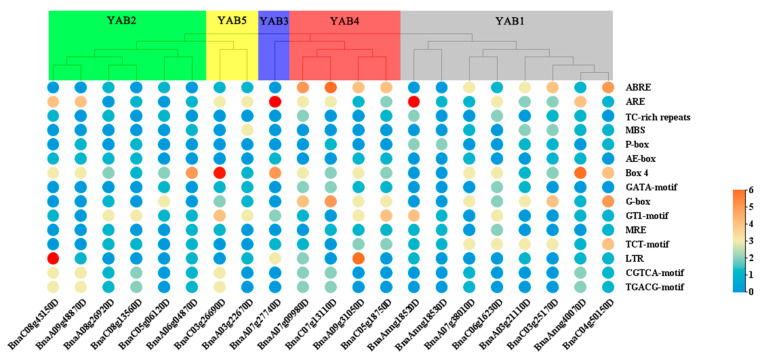
Analysis of major cis-acting elements in BnaYAB promoters. The number of cis-regulatory elements is indicated as a heatmap (Appendix A).

**Figure 6 genes-12-00981-f006:**
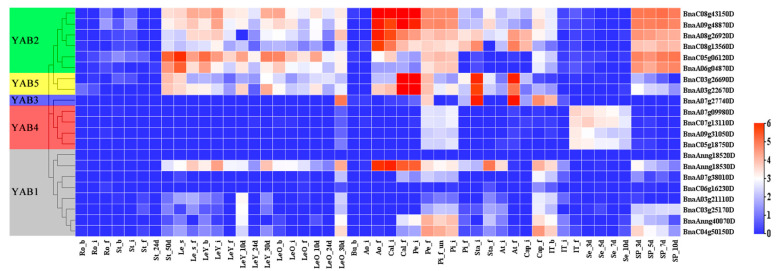
Heatmap representation of the expression patterns of BnaYAB genes in the ZS11 cultivar. The abbreviations below the heatmap indicate the different tissues and organs/developmental stages (Appendix A) relative expression is shown as a heatmap (Appendix A).

**Figure 7 genes-12-00981-f007:**
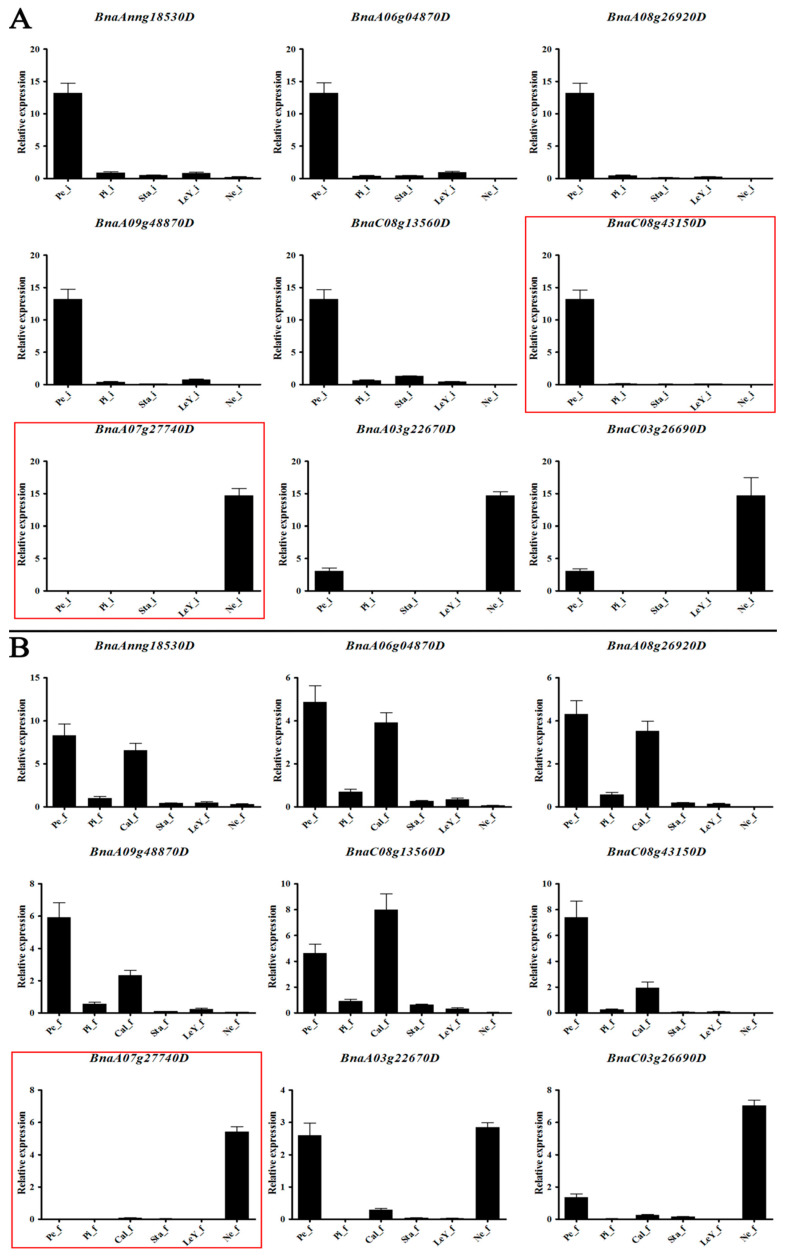
Relative expression of nine BnaYAB genes in different tissues and growth stages of *B. napus*. Relative expression levels in different tissues at the initial flowering stage (**A**), at the full flowering stage (**B**), and during the green pod stage (**C**). The abbreviations along the x-axes represent the different samples (Appendix A). A red box indicates that the gene is only highly expressed in one tissue.

**Table 1 genes-12-00981-t001:** Numbers of YABBY genes in Arabidopsis thaliana and five Brassica species.

Type	*A. thaliana*	*B. rapa*	*B. oleracea*	*B. nigra*	*B. juncea*	*B. napus*
*YAB1*	2	3	4	4	8	8
*YAB2*	1	3	3	3	3	6
*YAB3*	1	1	1	2	2	1
*YAB4*	1	2	2	2	4	4
*YAB5*	1	1	1	1	2	2

## Data Availability

Data used in this study are presented in the Appendix A.

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
