# Peer review of "Genome-Wide Analysis of the YABBY Transcription Factor Family in Rapeseed (Brassica napus L.)"

_genes, 2021, doi:10.3390/genes12070981_

Round 1

Reviewer 1 Report

Dear Author, 

         My major comment for this manuscript is how you identified YABBY gene in Brassica species. 

Previously we have used YABBY domain (PF04690) and also blast using Arabidopsis YABBY genes. Also need to mention which genome version author used for identification. 

Other comments can be found in attached revised manuscript version.

Author Response

Dear reviewer

Thank you very much for your suggestions, which helped us a lot

In response to your comments, we have made the following changes

  1. About the classification of YABBY family: For the convenience of follow-up research, phylogenetic analysis further classifies the six subgroups of YABBY protein, YAB1(FIL+YAB3), YAB2, YAB3(CRC), YAB4(INO), YAB5ï¼›
  2. Screening for homologous genes: This is our mistake. This article has performed a Blastp comparison of the Arabidopsis YABBY protein sequence, and the threshold used is E-value <1 × 10-20ï¼›
  3. The threshold we use for MCScanX is E-value <1 × 10-10ï¼›
  4. We are using Brapa_genome_v3.0, and 10 members have been identified in B.rapa, and at the same time, our Blastp comparison, the threshold used is E-value <1 × 10-20ï¼›

5.Further description of collinearity analysis, we have shown in the revised manuscriptï¼›

  1. Regarding the study of conserved domains, we think that Figure 4B is a supplement to motif, which further illustrates that some domains are particularly conserved in the same family, and the existence of YABBY domains is also when we identify YABBY members Conditions, we once again confirmed the MEME results, which are consistent with our stated resultsï¼›
  2. Some descriptions in section 3.4 of the article have been adjusted to the materials and methods sectionï¼›
  3. The last sentence of section 3.6 has been deleted accordinglyï¼›
  4. We have adjusted some sentences in the manuscript to make them more reasonable.

Thank you and best regards.

Yours sincerely,

Jichun Xia

Reviewer 2 Report

Dear Authors!

The manuscript entitled 'Genome-Wide Analysis of the YABBY Transcription FactorFamily in Rapeseed (Brassica napus L.)' provides novel and important information on the YABBY family of plant-specific transcription factors in Brassica species. It is an interesting and comprehensive analysis of the gene family in 5 chosen Brassica species. The introduction is nicely written, gives all the necessary information to properly understand the content of the manuscript. Materials and methods are presented in a clear and precise manner. Results are presented in an understandable fashion, supported by Table, Figures and Supplementary Data. Discussion is properly written and satisfactory. I have some minor comments and a few question, that are given in attached file.

Author Response

Dear reviewer

Thank you very much for your suggestions, which helped us a lot

In response to your comments, we have made the following changes

  1. We have modified the punctuation and formatting in the article;
  2. In the collinearity analysis, we only analyzed Arabidopsis, cabbage, cabbage and Brassica napus. This place is our writing error, it should be Arabidopsis and three Brassica crops;
  3. The red box indicates that the gene is only highly expressed in one tissue, and we have made corresponding supplements in the article;
  4. In the analysis of cis-acting elements, we believe that the existence of cis-regulatory elements makes the regulation of gene families specific.For example, the presence of a large number of light-responsive elements indicates that the expression of YABBY gene in Brassica napus may be strictly controlled by photosynthesis.Studies have shown that this family has a regulatory role in the development of plant leaves, and we speculate that it may further play a role by affecting the development of chloroplasts.

In addition to light response elements, we have also discovered hormones, indicating that this family has many effects on plant development, and the specific pathways of action need to be further verified;

  1. For some word orders and unreasonable words, we also make corresponding changes.

Thank you and best regards.

Yours sincerely,

Jichun Xia